# Epigenomics Analysis of the Suppression Role of *SIRT1* via H3K9 Deacetylation in Preadipocyte Differentiation

**DOI:** 10.3390/ijms241411281

**Published:** 2023-07-10

**Authors:** Youzhualamu Yang, Wei Peng, Xiaolong Su, Binglin Yue, Shi Shu, Jikun Wang, Changqi Fu, Jincheng Zhong, Hui Wang

**Affiliations:** 1Key Laboratory of Qinghai-Tibetan Plateau Animal Genetic Resource Reservation and Utilization, Sichuan Province and Ministry of Education, Southwest Minzu University, Chengdu 610225, China; yangyouzlm@163.com (Y.Y.); 15682844651@163.com (X.S.); yuebinglin123@163.com (B.Y.); xdzwjk@163.com (J.W.); zhongjincheng518@126.com (J.Z.); 2Qinghai Academy of Animal Science and Veterinary Medicine, Qinghai University, Xining 810016, China; pengwei0112@sohu.com (W.P.); shushi0211@163.com (S.S.); fuchangqimau@163.com (C.F.)

**Keywords:** SIRT1, intramuscular fat content, H3K9, acetylation, yak

## Abstract

Sirtuin 1 (*SIRT1*) overexpression significantly inhibits lipid deposition during yak intramuscular
preadipocyte (YIMA) differentiation; however, the regulatory mechanism remains unknown. We elucidated the role of
*SIRT1* in YIMA differentiation using lentivirus-mediated downregulation technology and conducted
mRNA-seq and ChIP-seq assays using H3K9ac antibodies after *SIRT1* overexpression in order to reveal
*SIRT1* targets during YIMA adipogenesis. Gene Ontology (GO) and Kyoto Encyclopedia of Genes and Genomes (KEGG)
analyses were performed in order to identify the functional annotation of common genes. In addition, a potential target of
*SIRT1* was selected to verify its effects on the differentiation and proliferation of YIMAs.
*SIRT1* interfered with lipid deposition and promoted YIMA differentiation. In total, 143,518 specific peaks
were identified after *SIRT1* overexpression, where genes associated with downregulation peaks were enriched
in transcription, gene expression, lipid-related processes, and classical lipid-related pathways. The H3K9ac signal in the
whole genome promoter region (2 kb upstream and downstream of the transcription
start site (TSS)) was weakened, and the peaks were distributed across all gene functional regions. Genes that lost signals in their
TSS region or gene body region were enriched in both biological processes and pathways associated with lipogenesis.
The ChIP-seq results revealed 714 common differential genes in mRNA-seq, which were enriched in “MAPK signaling”,
“lipid and atherosclerosis”, “mTOR signaling”, and “FoxO signaling” pathways.
A total of 445 genes were downregulated in both their H3K9ac signals and mRNA expression, and one of their most significantly
enriched pathways was FoxO signaling. Nine genes (*FBP2*, *FPGT*, *HSD17B11*,
*KCNJ15*, *MAP3K20*, *SLC5A3*, *TRIM23*, *ZCCHC10*,
and *ZMYM1*) lost the H3K9ac signal in their TSS regions and had low mRNA expression, and three genes
(*KCNJ15*, *TGM3*, and *TRIM54*) had low expression but lost their H3K9ac
signal in the gene body region. The interference of *TRIM23* significantly inhibited fat deposition during
preadipocyte differentiation and promoted cell proliferation by increasing S-phase cell numbers. The present study provides
new insights into the molecular mechanism of intramuscular fat content deposition and the epigenetic role of *SIRT1*
in adipocyte differentiation.

## 1. Introduction

Yak (*Bos grunniens*) is a special breed of livestock that is mainly distributed in the alpine pastoral areas of the Qinghai–Tibet Plateau in China and is the third largest cattle breed in China after the yellow cattle and buffalo. The extreme ecological conditions and long-term natural selection in its distribution area have led the yak to be uniquely resistant to low temperatures, low oxygen, and strong ultraviolet radiation compared to cattle and buffalo. Yak meat is popular among consumers because it is rich in protein and low in fat. However, thick muscle fibers and poor tenderness are two major bottlenecks that restrict the development of the yak industry [1]. Intramuscular fat (IMF) is fat deposited by adipocytes into the connective tissues surrounding muscle fibers, and directly affects the sensory quality of the muscle, positively correlated with muscle tenderness, juiciness, and flavor [2]. IMF deposition has attracted widespread attention over the years. Genetic, management, and nutritional factors are the main factors that affect IMF deposition [3]; however, genetic factors additionally present moderate to more pronounced effects [4].

Sirtuin 1 (SIRT1), a nicotinamide adenine dinucleotide (NAD)-dependent enzyme, is a class III histone deacetylase (HDAC). *SIRT1* is involved in a variety of physiological and pathological processes, such as cell proliferation and apoptosis, cellular senescence, and glucose and lipid metabolism [5,6,7]. *SIRT1* triggers lipolysis and induces fat mobilization by repressing the docking of PPARγ and its cofactors (nuclear receptor corepressor, silencing mediator of retinoid, and thyroid hormone receptors), and the *SIRT1* activator (resveratrol) inhibits adipocyte differentiation by downregulating PPARγ activity [8]. *SIRT1* inhibits FoxO1 transcriptional activity and affects lipid metabolism by regulating the deacetylation levels of FoxO1 and promoting its binding to *PPARγ* and *C/EBPα* [9]. In vivo and in vitro experiments have shown that *SIRT1* plays an important role in the reduction of excessive lipid accumulation by lutein in mice [10] and influences lipocalin-mediated lipid synthesis in keratinocytes to maintain skin barrier homeostasis [11]. Other studies in cattle have shown that the SNP locus on the 3′ flanking region of the *SIRT1* gene is significantly associated with IMF content [12]. Our previous studies using adenovirus-mediated overexpression technology have also revealed that *SIRT1* has a negative effect on IMF deposition [13]. However, whether *SIRT1* can regulate gene expression through H3K9 acetylation remains unclear.

In the present study, we elucidated the role of *SIRT1* during preadipocyte differentiation using lentivirus-mediated knockdown technology, and performed a ChIP-seq assay using H3K9ac antibodies and mRNA-seq after *SIRT1* overexpression in order to identify large-scale gene expression changes that may be epigenetic modifications caused by *SIRT1*. In addition, we selected *TRIM23* (a potential target of *SIRT1* deacetylation modification) for further analysis in order to determine its role during preadipocyte differentiation.

## 2. Results

### 2.1. Effect of SIRT1 Interference on Fat Deposition in Yak Preadipocytes

Figure 1A shows a lentivirus-mediated *SIRT1*-shRNA interference system and infected YIMAs. RT-qPCR results showed that *SITR1* mRNA levels reduced by approximately 73.5% after 48 h of differentiation. The downregulation of *SIRT1* significantly (*p* < 0.05) increased the expression of lipid differentiation marker genes (*PPARγ*, *C/EBPα*, and *AP2*), de novo lipid synthesis-related genes (*FASN* and *ACACA*), the fatty acid transport-related gene (*FABP3*), stearoyl-coenzyme A desaturase (*SCD*), *SREBP1*, *SCAP*, and the lipid droplet-related gene (*PLIN2*). However, the expression of the lipolysis-related gene (*LPL*) and fatty acid oxidation-related genes (*CPT1A* and *PGC*-*1α*) decreased significantly (*p* < 0.05; Figure 1B–D). *SIRT1* interference facilitated the adipogenesis of YIMAs, which was confirmed by the increased number of lipid droplets, TAG accumulation in oil red O staining (Figure 1E,F), and TAG content assays performed after 48 h (Figure 1G). Based on our previous results, we conclude that *SIRT1* negatively regulates TAG and lipid droplet accumulation.

### 2.2. Genome-Wide Changes Underlying H3K9 Deacetylation after SIRT1 Overexpression

A total of 143,518 peaks in the *SIRT1* overexpression group and 109,126 peaks in the control group were identified using a ChIP-seq assay, with 178,583 common peaks overlapping (by at least 1 bp) between the two groups (Figure 2A). The genome-wide distribution of the histone mark H3K9ac showed that the signal was downregulated near the TSS region (Figure 2B). Notably, in addition to the promoter regions, differential peaks were distributed across all functional regions of the genes (Figure 2C). The GO analysis of the downregulated genes showed that they were enriched in terms associated with the transcription, gene expression, and lipid-related processes, including “regulation of gene expression”, “regulation of transcription by RNA polymerase II” “response to lipid”, and “regulation of fat cell differentiation” (Appendix A).

The pathway analysis of the downregulated genes revealed enrichment in certain lipid metabolism-related pathways, such as the PI3K-Akt signaling pathway, insulin resistance, and HIF-1 signaling pathway (Appendix A). The genes were divided into two groups based on the ChIP signal: (1) gained and (2) lost H3K9ac signal (Figure 2D). The GO and KEGG pathway analyses results showed that the two groups of genes were enriched in biological processes and signaling pathways, including “lipid catabolic processes”, “medium-chain fatty acid metabolic processes”, “fatty acid biosynthetic processes”, and “medium-chain fatty acid biosynthetic processes”. Furthermore, genes that lost the ChIP signal near the TSS region were enriched in cell division-related pathways, such as “cell cycle checkpoint” and “DNA synthesis” (Figure 3A–D). Some key lipid-related genes that are involved in the loss of acetylation signals in the distal intergenic (*FOXO1* and *PGC*-*1α*), intron (*HIF1A*), and promoter (*UCP2*) regions were identified (Figure 3E). The ChIP assay results were consistent with those of the ChIP-seq (Figure 3F), which confirmed that the identification of genes associated with the differential peaks in this study was reliable. Collectively, the results suggest that *SIRT1* regulates gene expression through the deacetylation of H3K9ac in the TSS and gene body regions.

### 2.3. Association between ChIP-seq and RNA-seq Data

The integrated analysis of ChIP-seq and RNA-seq data showed that there were 714 common genes (Figure 4A). KEGG pathway analysis revealed that the genes were involved in some classical lipid-related pathways, such as the “MAPK signaling”, “lipid and atherosclerosis”, “mTOR signaling, and “FoxO signaling” pathways (Appendix A).

A total of 445 common differential genes with simultaneously reduced H3K9ac signals and mRNA expression were identified (Figure 4B). KEGG pathway enrichment analysis showed that the genes were enriched in lipid metabolism-related signaling pathways, such as the “FoxO signaling pathway” (Figure 4C). These results demonstrate that *SIRT1* overexpression downregulated lipid-related gene expression through changes in H3K9ac acetylation signals and *SIRT1* influences lipogenesis by negatively modulating lipid deposition in yak preadipocytes. Several H3K9ac signals were lost in the TSS and gene body regions after *SIRT1* overexpression, suggesting that the regulation of target lipid-related genes by *SIRT1* not only occurred in the promoter region but also in the gene body regions. In addition, H3K9ac signals of nine genes *(FBP2*, *FPGT*, *HSD17B11*, *KCNJ15*, *MAP3K20*, *SLC5A3*, *TRIM23*, *ZCCHC10*, and *ZMYM1*) were lost in the TSS region and their mRNA expression was downregulated, while the H3K9ac signals of three genes (*KCNJ15*, *TGM3*, and *TRIM54*) were lost in the gene body region and their mRNA expression was downregulated (Figure 4D,E).

### 2.4. TRIM23 Knockdown Suppresses Adipocyte Differentiation and Proliferation

*TRIM23* has been reported to regulate PPARγ atypical polyubiquitination and proteasomal degradation, further influencing adipocyte adipogenesis [14]. In our above results, we found that the H3K9ac signal of *TRIM23* in its TSS region was decreased, accompanied by a decrease in mRNA levels after *SIRT1* overexpression (Figure 4D). So, we further designed experiments to determine its function in adipocyte differentiation and proliferation. RT-qPCR analysis revealed that *TRIM23* was expressed in preadipocytes and that its mRNA levels increased during adipogenesis (Figure 5A). si*TRIM23* effectively reduced *TRIM23* mRNA levels by 58% when compared to the control group (Figure 5B). To test whether *TRIM23* affects lipid metabolism in YIMAs, we investigated the expression of adipocyte differentiation marker gene (*PPARG*), genes involved in de novo fatty acid synthesis (*SCD* and *SREBF1*), and a gene involved in TAG synthesis (*DGAT2*). *TRIM23* knockdown significantly decreased the expression of these genes (*p* < 0.05). With regard to the expression of fatty acid oxidation-related genes (*CPT1A* and *PGC*-*1α*), *PGC*-*1α* mRNA levels increased markedly after *TRIM23* knockdown (*p* < 0.01; Figure 5C–E). Similarly, *TRIM23* knockdown significantly reduced intracellular lipid droplet deposition (Figure 5F–H) and TAG content (Figure 5I). We observed that *TRIM23* knockdown significantly upregulated the mRNA expression levels of proliferation-related genes (*CCND1* and *CCNE1*; *p* < 0.01; Figure 6A). Moreover, the scratch test and flow cytometry results showed that *TRIM23* knockdown increased cell proliferation (Figure 6B) and the number of S-phase cells but reduced the number of cells in the G0/G1 phase (Figure 6C). In summary, these results indicate that *TRIM23* knockdown inhibited IMF deposition but promoted cell proliferation in YIMAs.

## 3. Discussion

A relatively low IMF content in humans is better because IMF deposition can lead to obesity, which in turn, leads to a range of chronic diseases, such as type II diabetes and nonalcoholic fatty liver disease [15]. However, in livestock, IMF content is considered a key factor influencing meat quality and is positively correlated with meat tenderness, juiciness, and flavor. IMF content depends on the number of intramuscular adipocytes and their ability to synthesize fat. The processes of adipocyte division, differentiation, and adipose synthesis are controlled by a complex network of transcription factors [16]. Furthermore, IMF is a highly heritable trait [17]. Therefore, elucidating the regulatory mechanisms of such transcription factors and enzymes in YIMAs is of great socioeconomic importance in the improvement of meat quality and breeding superior breeds.

SIRT1 (one of the seven mammalian sirtuins) is a NAD^+^-dependent deacetylase that regulates adipogenesis in vitro and in vivo by removing acetyl groups from many histone and non-histone proteins (transcription factors and transcriptional coregulators) [18]. Fat accumulation in 3T3-L1 cells was reduced after *SIRT1* overexpression, while TAG content was significantly increased after *SIRT1* inhibition, indicating that *SIRT1* acts as a negative modulator of adipogenesis in mice [8]. Similarly, in our previous studies, *SIRT1* overexpression decreased TAG content and lipid droplet deposition. In contrast, in this study, we observed that *SIRT1* expression in YIMAs increased the expression of adipocyte differentiation marker genes and other lipid synthesis-related genes, leading to lipid droplet deposition. Overall, we concluded that *SIRT1* negatively regulates lipid synthesis in YIMAs. 

Transcriptional and epigenetic mechanisms work together to regulate gene expression, and epigenetics has become an important research area in recent years. Several histone-modifying enzymes regulate adipogenesis by producing active histone codes, such as acetylated H3K9 and methylated H3K4, or repressive codes, such as methylated H3K9 and H3K27. Examples of such enzymes include histone acetyltransferases (targeting histone H3K9) that promote adipogenesis and histone deacetylases that repress adipogenesis [19]. *SIRT1* represses gene expression by silencing the chromatin structure through its histone deacetylation activity [20]. Studies on senescent cells have shown that *SIRT1* inhibits the expression of SASP factors through histone deacetylation in its promoter region and is associated with the histone deacetylation of H1 lysine 26 (H1K26ac), H3 lysine 9 (H3K9ac), and H4 lysine 16 (H4K16ac) [21]. *SIRT1* regulates lipid homeostasis by deacetylating transcription factors, including FoxOs, PGC−1a, and PPARγ [22,23,24]. In addition, the epigenetic regulation of *SIRT1* may affect lipogenesis by deacetylating H3K9ac and H3K14ac signals of lipogenesis-related genes, such as *HIF1A*, *ChREBP*, and *PNPLA3* [25,26,27]. Therefore, to verify the epigenetic regulation mechanism of *SIRT1* through histone acetylation levels, we performed ChIP-seq analysis using an H3K9ac antibody in yak intramuscular adipocytes after *SIRT1* overexpression.

Genome-wide distribution analysis revealed that the H3K9ac signal was downregulated near the TSS region. Notably, we observed peaks distributed across all functional regions of the genes, confirming that gene expression is regulated via the coordination between proximal and distal chromatin elements [28]. Previous studies have shown that *SIRT1* knockdown increases H3K14ac levels in *HIF1A*, thereby increasing its expression levels [25]. However, *HIF1A*, a gene that influences intracellular lipid accumulation under hypoxic conditions [29], decreased H3K9ac levels after *SIRT1* overexpression. The results of previous studies and those of the current study suggest that the mechanism of *SIRT1* in *HIF1A* epigenetic regulation is the simultaneous deacetylation of H3K9ac and H3K14ac. In addition, among the genes with decreased H3K9ac signal, *UCP2*, *FOXO1*, *PGC-1α*, and other genes have been shown to be involved in lipogenesis [30,31,32].

The analysis of the genes with varying H3K9ac signals in the TSS and gene body regions revealed that genes that lost the ChIP signal in the TSS region were enriched in cytokinesis-related pathways, indicating that *SIRT1* overexpression negatively regulates yak intramuscular adipocyte cytokinesis through histone H3K9ac deacetylation in the TSS region. However, the observation is inconsistent with the findings of previous studies showing that *SIRT1* overexpression increases the proliferative ability of bone marrow-derived macrophage during differentiation [33], which may be due to the different cell types or regulation mechanisms of *SIRT1*. In addition, H3K9ac signal enrichment analysis showed that genes that lost the signal in the gene body region were enriched in biological processes and signaling pathways associated with adipogenesis. Nine genes (*FBP2*, *FPGT*, *HSD17B11*, *KCNJ15*, *MAP3K20*, *SLC5A3*, *TRIM23*, *ZCCHC10*, and *ZMYM1*) and three genes (*KCNJ15*, *TGM3*, and *TRIM54*) exhibited decreased H3K9ac signals in the TSS or gene body region, in addition to decreased mRNA expression levels associated with lipogenesis or lipid metabolism, and *FBP2* is involved in gluconeogenesis in mammalian skeletal muscles [34]. *HSD17B11* regulates lipid droplet dynamics and lipid metabolism by altering *ATGL* expression [35]. In hepatocellular carcinoma cells, *TRIM54* overexpression results in the significant activation of the Wnt/b-catenin pathway [36]. *TRIM23* is a novel positive regulator of adipocyte maturation that regulates the abundance of PPARγ to control the conversion of early to late adipogenic enhanceosomes [14]. PPARγ is a critical transcription factor of adipogenesis and a potent modulator of preadipocyte differentiation and lipid metabolism [37]. Various studies have shown that *SIRT1* regulates important adipocyte-specific genes in a histone acetylation-dependent manner, including adipogenesis, lipolysis and lipogenesis. For instance, *SIRT1* may regulate the expression of Paladin-like phospholipase domain 3 (*PNPLA3*)-containing genes through the acetylation modification of H3K9, which exhibits biphasic effects on lipid synthesis and lipolysis in vitro [27]. The reduced enrichment of SIRT1 in the *PPAR-γ* promoter region resulted in increased levels of histone acetylation and promoted *PPAR-γ* expression [38]; correspondingly, *SIRT1* causes the histone deacetylation of the *UCP3* promoter through its deacetylase activity, thereby suppressing *UCP3* expression [39]. In mice liver, *CD36* expression can be promoted by reducing SIRT1 binding to histone at *CD36* promoter H3, thereby regulating hepatic triglyceride accumulation and also stimulating the activity of hepatic SIRT1 to histone H3-Lys9 deacetylation, which in turn leads to a decrease in mature nuclear SREBP-1 and thus prevents lipid accumulation [40,41]. In previous studies, we also found that *SIRT1* regulates certain transcription factors and genes in a histone acetylation-dependent manner to affect IMF deposition in yak [13]. Combined with the results of the present study, we suspected that *SIRT1* regulates IMF deposition in yak through deacetylation at the histone H3 of these target genes, and these target genes selected in this study can also be used as a direction for future studies.

Although we successfully found that *SIRT1* inhibited the expression of adipogenic and cell division genes by altering H3K9ac signals in the TSS or gene body region, and interference *TRIM23* influences IMF deposition and adipocyte proliferation, some limitations in this study should be noted. Since SIRT1 and TRIM23 antibodies to yak are not commercially available, the protein levels were not detected in our research. Nonetheless, with the functional analysis of *SIRT1* and the combined analysis of CHIP-seq and RNA-seq, many epigenetic targets of *SIRT1* that are involved in lipid-related metabolism pathways may be associated with IMF content, and these possibilities are worthy of further research efforts.

## 4. Materials and Methods

### 4.1. Cell Culture and Treatment

YIMAs were isolated from longissimus dorsi muscle tissues according to previously published protocols [42]. Briefly, yaks were humanely sacrificed, longissimus dorsi tissues were excised immediately between the 12th and 13th ribs (right half carcass) and rapidly washed with PBS supplemented with antibiotics (3% penicillin/streptomycin), and immediately transported to the laboratory. The tissues were immediately chopped into small pieces of approximately 1 mm3, and then digested with 2.5 mg/mL type II collagenase as the digestion buffer (Sigma-Aldrich, St. Louis, MO, USA) in a water bath at 37 °C for 2 h, followed by washing with Hank’s balanced salt solution (Thermo Fisher Scientific, Waltham, MA, USA). The digested tissues were filtered using 70 μM and then 40 μM filters and centrifuged at 1800 rpm × min^-1^ for 5 min to remove the supernatant, and the residual lower fraction was used for adipocyte identification, as in previous studies [43]. The cells were washed with DMEM (Thermo Fisher Scientific, Waltham, MA, USA) and centrifuged twice at 1500 rpm × min^−1^ for 5 min. Thereafter, cells were rinsed with a growth medium without penicillin/streptomycin and incubated at 37 °C in 5% CO_2_. After 1 h of incubation, the medium was replaced with a growth medium containing 90% basal DMEM/F12 medium, 10% fetal bovine serum, and 1% penicillin/streptomycin (100 U/mL).

### 4.2. Interfering Lentiviral Packaging, siRNA Interference, and Cell Transfection

The interfering vector pHBLV-U6-MCS-EF1-mcherry-T2A-PURO was selected for lentiviral packaging, and the *SIRT1* shRNA (*SIRT1*-shRNA) and control (NC-shRNA) sequences were designed according to the CDS region of the *SIRT1* gene (XM_024986766.1), as shown in Appendix A. Briefly, shRNA sequences were annealed and ligated with *EcoRI* and *BamHI* restriction enzymes. The interfering vector was ligated, and the correctly sequenced plasmids were transfected into 293T cells. The supernatant was collected at 72 h, and lentiviruses at 300 MOI were used to inject the cells for *SIRT1* interference. *SIRT1* overexpression (Ad_S) and its control, Ad-green fluorescent protein (Ad_G) adenovirus, were utilized, as in previous studies [13], and stored in our laboratory. *TRIM23* siRNA (si*TRIM23*) and its negative control (NC) were designed and synthesized by GenePharma Ltd. (Shanghai, China; Appendix A). In addition, 120 nM si*TRIM23* was transfected into cells for 6 h using Lipofectamine 3000 reagent (Invitrogen, Carlsbad, CA, USA). Before transfection, YIMAs were plated in 6-well plates at a density of 2 × 10^5^ cells for quantitative real-time PCR (RT-qPCR) and Oil Red O staining. For triacylglycerol (TAG) content determination, ChIP-seq, and mRNA-seq, cells were plated in 60 mm culture dishes at a density of 1 × 10^6^ cells. The cells for proliferation analysis were plated in 12-well plates at a density of 4 × 10^5^ cells. The cells were starved in growth medium without fetal bovine serum for 4 h prior to transfection or injection, after 6 h of transfection, the medium was replaced with a differentiation-inducing medium (growth medium containing 50 µM oleic acid) and the transfected cells were collected after 48 h for subsequent analysis.

### 4.3. Oil Red O Staining, Triglyceride Assay, and BODIPY Staining

Cells were harvested for Oil Red O Staining and cellular TAG content determination according to previous studies [44]. Intracellular Oil Red O stain was extracted using isopropanol and quantified by measuring the optical absorbance at 510 nm. For staining, the cells were first washed three times with PBS, fixed in 4% paraformaldehyde for 1 h at 25 °C, washed again with PBS, and finally stained with 2 μg/mL BODIPY 493/503 (D3922; Thermo Fisher Scientific, Waltham, MA, USA) or Oil Red O for 30 min at 25 °C. The cells were counterstained with DAPI after washing with PBS and photographed using an Axio Observer 3 (Zeiss, Oberkochen, Germany). Images of the control and treated cells were captured using default parameters.

### 4.4. RNA Extraction and RT-qPCR

Total RNA was extracted from cultured cells using TRIzol reagent (Invitrogen, Carlsbad, CA, USA), according to the manufacturer’s instructions. Total RNA (250 ng) was reverse transcribed to generate cDNA using the Prime Script RT reagent kit (RR047A; Takara Bio, Shiga, Japan), according to the manufacturer’s protocol. RT-qPCR was performed using SYBR Green Premix Ex Taq reaction mix with gene-specific primers. Gene-specific primers used in this study are listed in Appendix A.

### 4.5. ChIP Assays, High-Throughput Sequencing, and ChIP-qPCR

Cells were harvested after incubation with Ad_S or Ad_G for 48 h. ChIP assays were performed using H3K9ac (ab4441; Abcam, Cambridge, UK), as described in previous studies [13]. Briefly, cells were cross-linked with 1% formaldehyde (Roche, Basel, Switzerland) for 5 min, and cross-linking was terminated by adding 1 mL of 2 M glycine, followed through incubation for 5 min at 20 °C. Afterward, chromatin was sheared using an M220 Focused ultrasonicator (Covaris, Woburn, MA, USA) to obtain DNA fragments of approximately 500 bp. Thereafter, 5 µg of H3K9ac antibody was used to pull down the tagged proteins. The chromatin was then de-crosslinked by adding proteinase K and incubating overnight at 65 °C, and the DNA was purified using MinElute PCR purification kit (Qiagen, Hilden, Germany). Subsequently, high-throughput sequencing libraries for ChIP-DNA and all mRNAs of the cultured cells were constructed using an Illumina Hiseq 2500 sequencing platform (Novogene Biotech Co., Ltd., Beijing, China). All raw data were deposited at the National Center for Biotechnology Information (NCBI) Gene Expression Omnibus repository under accession number PRJNA992635.

The raw data were aligned with the yak reference genome (BosGru v2.0), as described in previous studies [13]. ChIPseeker [45] was used to retrieve the nearest genes to the peaks and annotate the genomic region of each peak. GO enrichment analysis was then performed using the GOseq R package to identify the functional enrichment results. GO terms with a corrected *p*-value of <0.05 were considered significantly enriched in peak-related genes. We used the KOBAS software to test the statistical enrichment of peak-related genes in the KEGG pathways. Based on the genes corresponding to the reads, the signal was lost or gained in the TSS and gene body regions. Reads uniquely matched to the genomic loci were selected for functional and enrichment analyses.

The immunoprecipitated DNA was used for ChIP-qPCR using the primers listed in Appendix A. Fold enrichment was then calculated as: percentage input = 100 × 2 (CT input sample—CT immunoprecipitation sample) and CT = threshold cycle of PCR, followed by normalization to the indicated controls.

### 4.6. Analysis of Associations between ChIP-seq and RNA-seq Data

We previously constructed six transcriptome libraries (NCBI accession number PRJNA852863) of cells treated with the same cells as those used in the current ChIP-seq. Here, we performed a correlation analysis between gene expression in the transcriptome and ChIP-seq data. First, the intersection of all the genes with different peak signals in the range of 20 kb and all differentially expressed genes in mRNA-seq data were identified. The enrichment analysis was then performed for the intersection genes that were downregulated. Finally, the intersection genes that gained or lost H3K9ac signal in the TSS or gene body region were subjected to enrichment analysis.

### 4.7. Proliferation Detection by Flow Cytometry

Sterile pipette tips were used to scratch along the central axis of the culture plates at a uniform speed, and scratch widths at different incubation times were recorded. Flow cytometry was performed on a Sysmex Cube 8.0 platform (Sysmex Corporation, Kobe, Japan) as previously described [13]. Briefly, the cells were washed with precooled PBS solution, centrifuged at 1500 rpm × min^−1^ for 5 min, and then fixed overnight in 75% absolute ethanol at 4 °C. PI reagent (50 ng/mL; Solarbio, Beijing, China) was added for incubated for 30 min at 20 °C after washing in PBS.

### 4.8. Statistical Analysis

All data are presented as means ± standard error of the mean. Student’s *t*-test or one-way ANOVA with Duncan’s test was performed when appropriate. All data were analyzed using GraphPad Prism 8 (GraphPad Software Inc., La Jolla, CA, USA), and the difference was statistically significant at *p* < 0.05; * and ** denote *p* < 0.05 and *p* < 0.01, respectively.

## 5. Conclusions

In conclusion, our study revealed that *SIRT1* is a negative regulator of IMF deposition in yaks. *SIRT1* alters the H3K9 acetylation levels of numerous genes in the TSS and gene body regions to regulate lipid metabolism, including nine (*FBP2*, *FPGT*, *HSD17B11*, *KCNJ15*, *MAP3K20*, *SLC5A3*, *TRIM23*, *ZCCHC10*, and *ZMYM1*) and three (*KCNJ15*, *TGM3*, and *TRIM54*) potential target genes regulated in the TSS or gene body region. Furthermore, *TRIM23* significantly inhibited fat deposition during preadipocyte differentiation. However, further studies are warranted to better understand the mechanisms underlying the “crosstalk” between *SIRT1* and genes with lost signals in the TSS and gene body regions, including *TRIM23*.

## Figures and Tables

**Figure 1 ijms-24-11281-f001:**
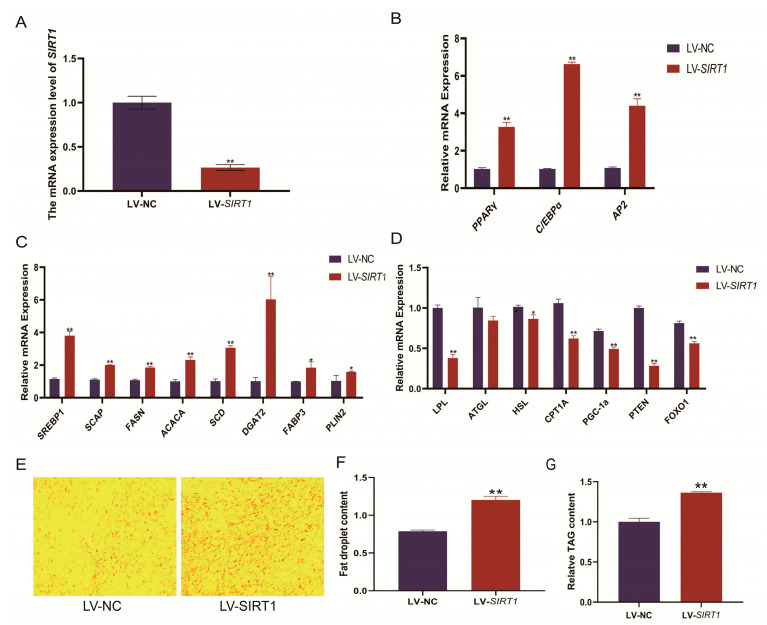
Effect of *SIRT1* interference on fat deposition in the preadipocytes of yak. (**A**) The interference efficiency of the *SIRT1* lentivirus. (**B**–**D**) Effects of *SIRT1* interference on the expression level of genes related to lipid metabolism: (**B**) markers during lipid differentiation; (**C**) de novo fatty acid synthesis and fatty acid transport-related genes; and (**D**) lipolysis and fatty acid oxidation-related genes. (**E**–**G**) Effects of *SIRT1* interference on YIMA adipogenesis: (**E**) Oil Red O staining; (**F**) lipid droplet accumulation; and (**G**) TAG content. All of the results were compared with the LV-NC group, were performed in triplicate, and were repeated three times (*n* = 6). Values are presented as mean ± SEM. * *p* < 0.05, ** *p* < 0.01.

**Figure 2 ijms-24-11281-f002:**
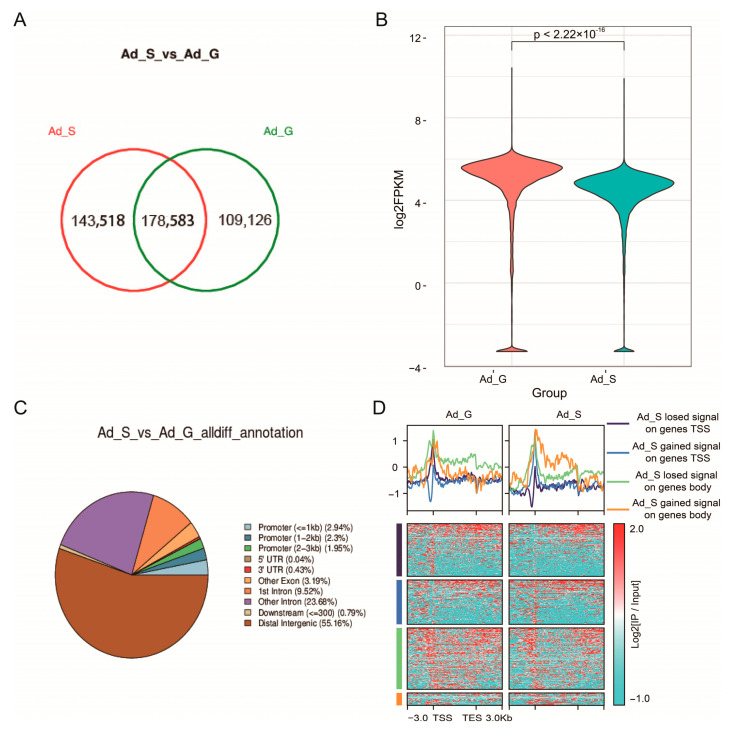
ChIP-seq analysis after SIRT1 overexpression. (**A**) The Venn diagrams of the peaks between Ad_G and Ad_S. (**B**) ChIP-seq signal levels in the promoter region (2 kb region upstream and downstream of the TSS) of genes. (**C**) Pie chart showing the distribution of peaks across the genome. (**D**) Heatmaps illustrate the best or unique peek lost or gained signals around genes. Each row represents a region of ±3 kb around the transcription start site (TSS) or transcription end site (TES). All the results were compared with Ad_G group and were performed in duplicate (*n* = 2; values are presented as mean ± SEM).

**Figure 3 ijms-24-11281-f003:**
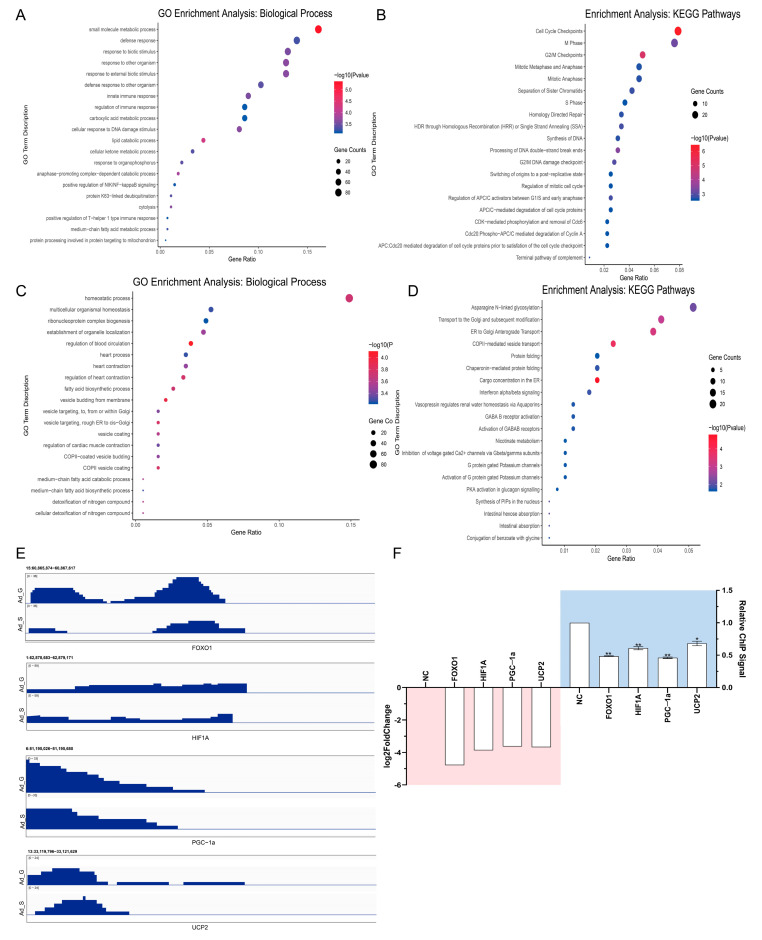
ChIP signal annotation and ChIP-qPCR result. (**A**,**B**) GO and KEGG pathways analysis in genes with unique ChIP signal lost in TSS. (**C**,**D**) GO and KEGG pathways analysis in genes with unique ChIP signal lost in the gene body. (**E**) Binding peaks at lipid-related genes (Integrative Genomics Viewer). (**F**) Decreasing peaks in lipid-related genes were analyzed via ChIP-qPCR. All the results were compared with Ad_G group. The results of (**A**–**E**) were performed in duplicate (*n* = 2), and the results of (**F**) were performed in triplicate and repeated three times (*n* = 6). Values are presented as mean ± SEM. * *p* < 0.05, ** *p* < 0.01.

**Figure 4 ijms-24-11281-f004:**
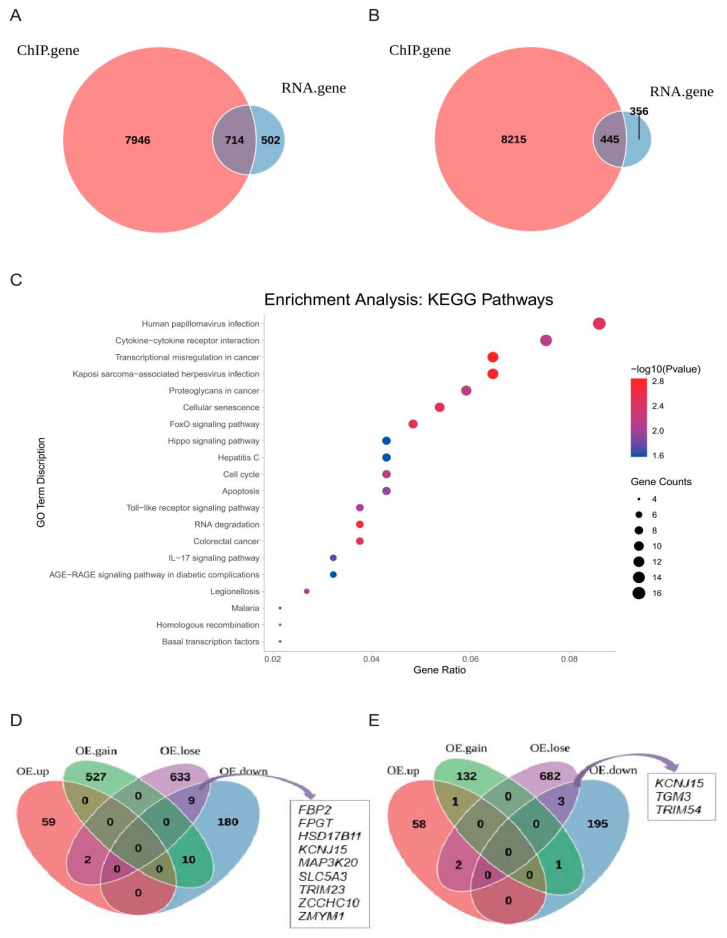
Combined ChIP-seq and mRNA-seq analysis after *SIRT1* overexpression. (**A**) The Venn diagrams of the genes within 20 kb around the differential peak in ChIP-seq with the differentially expressed genes in mRNA-seq (1.25-fold change). (**B**) Venn diagram of genes with reduced levels of both ChIP and mRNA expression. (**C**) KEGG pathways analysis in genes with reduced levels of both ChIP and mRNA expression. (**D**,**E**) Venn diagrams of genes that gained or lost ChIP signal at the TSS and gene body making intersections with genes up and downregulated at the mRNA level after *SIRT1* overexpression.

**Figure 5 ijms-24-11281-f005:**
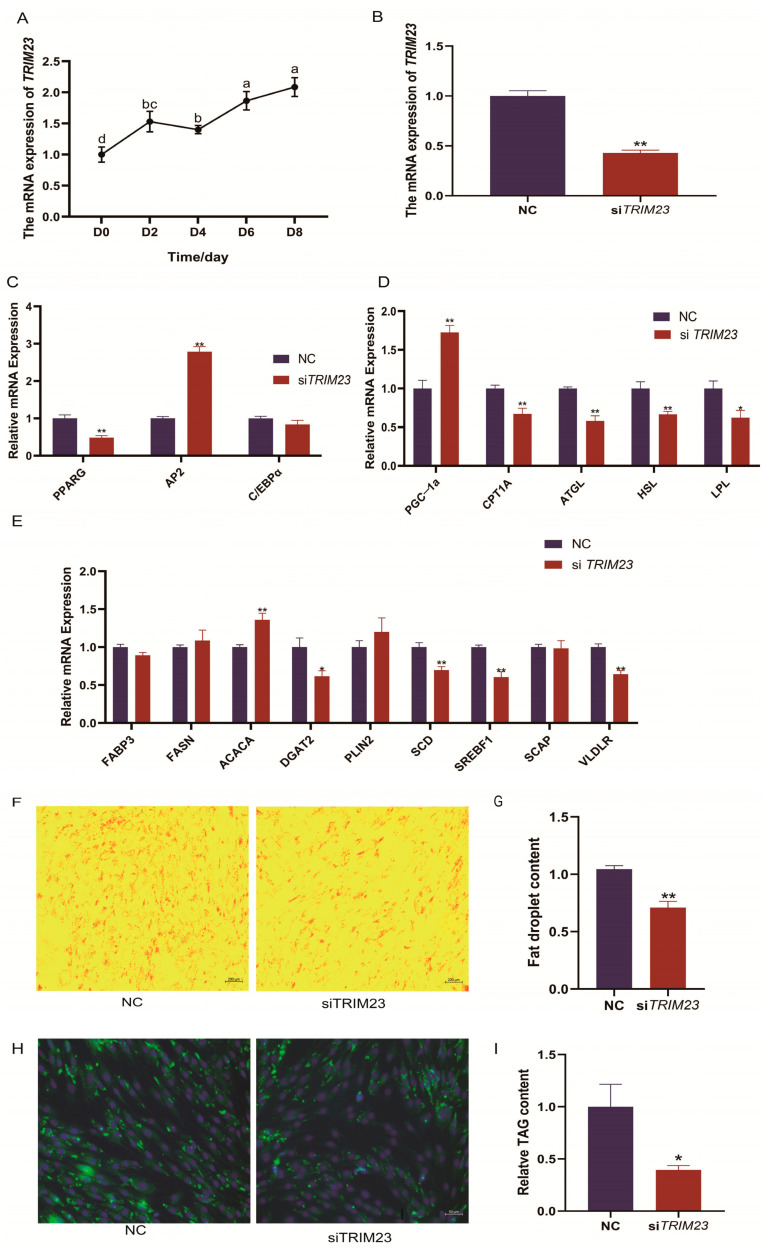
Knockdown of *TRIM23* inhibit yak intramuscular adipocyte lipid accumulation. (**A**) The mRNA expression of *TRIM23* during preadipocytes differentiation. Different lowercase letters (a, b, c, and d) in figures indicate significant differences among different groups during intramuscular adipogenesis, compared with the D0 group. (**B**) The relative mRNA expression level of *TRIM23* after si*TRIM23* interference. (**C**) The expression of *PPARγ*, *AP2*, and *C/EBPα* markers of lipid differentiation. (**D**) The expression of the lipolysis-related gene (*LPL*), *ATGL*, and *HSL*, Fatty acid oxidation-related genes (*CPT1A*, *PGC*-*1α*). (**E**) The expression of *SREBF1*, *SCAP*, lipid de novo synthesis-related genes (*FASN*, *ACACA*), *SCD*, *DGAT2*, fatty acid transport-related gene (*FABP3*), *PLIN2*, and *VLDLR*. (**F**,**G**) Oil Red O staining and lipid droplet content. (**H**) The body staining. (**I**) The triglyceride assays. The NC group was used as control group in (**B**–**I**), and all the results were performed in triplicate and repeated three times (*n* = 6). Values are presented as mean ± SEM. * *p* < 0.05, ** *p* < 0.01.

**Figure 6 ijms-24-11281-f006:**
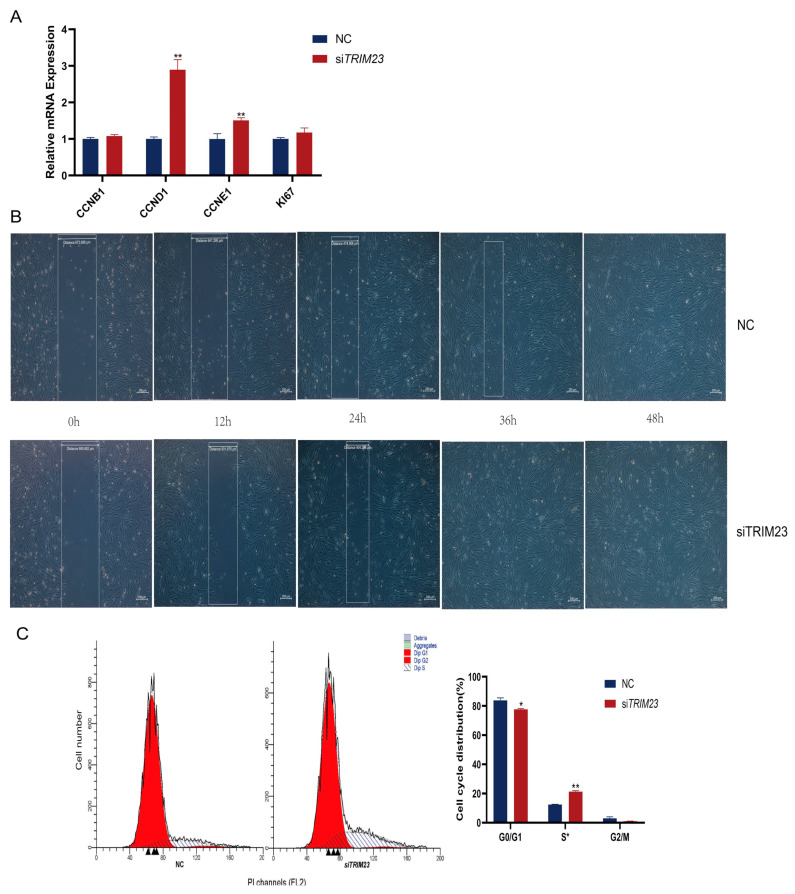
Knockdown of *TRIM23* promotes the proliferation of adipocytes in yak muscle. (**A**) The expression of proliferation-related genes (*CCNB1*, *CCND1*, *CCNE1*, *KI67*). (**B**) Scratch test. (**C**) Flow cytometry after *TRIM23* knockdown. The NC group was used as a control group, and all the results were performed in triplicate and repeated three times (*n* = 6). Values are presented as mean ± SEM. * *p* < 0.05, ** *p* < 0.01.

## Data Availability

All relevant data can be found within the paper and its Appendix A, and the raw RNA sequencing data generated during the current study have been uploaded to the NCBI BioProject with the accession number PRJNA644042.

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
