# Peer review of "Epigenomics Analysis of the Suppression Role of SIRT1 via H3K9 Deacetylation in Preadipocyte Differentiation"

_ijms, 2023, doi:10.3390/ijms241411281_

Round 1
Reviewer 1 Report
The concept and the methodology of the paper is sound and correct. Figure presentation is also clear, English language is appropriate. I have one minor issue only: the yak industry may not be as relevant in all parts of the world as it is for the Asian authors, although this topic selection is certainly novel, under-appreciated and has a sustainable potential. Still it would be nice to extend the introduction and the conclusion with comparison to other large breeds like cattle, buffalo etc. This would enhance visibility of the paper too that pays off in later citations.
Author Response
请参阅附件。

Reviewer 2 Report
The authors investigated the role and mechanisms of action of SIRT 1 in preadipocyte differentiation in yak muscle tissue. They found that Sirt 1 regulates intramuscular lipid accumulation by inhibiting the expression of cell division genes and adipogenic genes through altering H3K9ac acetylation signals. They further showed that TRIM23 significantly inhibited lipid deposition during preadipocyte differentiation and promoted cell proliferation.
Main comments:
1) A shortcoming of the study is that it is not possible to distinguish from which localization in muscle the preadipocytes were isolated given that localization determine different metabolic functions of adipocytes. Lipid accumulation in skeletal muscle is known to occur in several locations: a) intermuscular fat depots represent adipocytes localized between individual muscle, b) intramuscular lipid depots where adipocytes located between the muscle fiber, c) intramyocellulaf lipids droplets within the myocytes. Adipocytes located in a) plus b) are usually referred to as intermuscular adipocytes.
From which locations in the muscle were preadipocytes isolated?
And can they clearly be characterized as intramuscular?
2) Adipocytes in individual locations may originate from different cell sources (fibro-adipogenic progenitors, muscle satellite stem cell, etc.). Could the different origin of preadipocytes influence their gene expression during differentiation?
3) It is unclear whether preadipocytes were obtained from the stroma-vascular fraction and, if so, how they were separated from fibroblasts. Was the upper or lower fraction of isolated cells after centrifugation used for the study?
4) The Methods section does not describe how long isolated cells were incubated in growth medium before their further use for ChIP-seq and mRNA-seq analysis. Analysis of TRIM23 knockdown, monitoring of H3K9 deacetylation and more.
5) The development of preadipocytes to the stage of triglyceride accumulation takes several days. Provide details of how the cells were incubated and treated before measuring Oil Red staining, lipid droplet accumulation, and TAG content.
6) In the description of the Figures, it should be stated how many samples were analysed and also the statistical evaluation, i.e. which groups were compared to each other. Statistical significance (P < 0.05...) should be indicated in the legend below each Figure. It is insufficient to state the significance only in the "Statistical analysis" section.
7) Lines 202 - 203: The sentence " IMF content depends on the number of intramuscular adipocytes and their ability to synthesize adipose tissue.“ is incorrect. Adipocytes do not have the ability to synthesize adipose tissue. This needs to be corrected.
8) Some sentences are unnecessarily long and hard to understand. Such sentences should be modified. An example can be the sentence on lines 256 - 261.
Many sentence are not completely understable.
Reviewer 3 Report
In the current manuscript, Yang et al. claim that “Epigenomics analysis of SIRT1 suppression role via H3K9 deacetylation in preadipocytes differentiation”. The authors conclude that SIRT1-TRIM23 regulates Yak's preadipocyte differentiation and proliferation in a histone H3K9 acetylation-dependent manner. However, the conclusion that SIRT1 or its target TRIM23 affects adipogenesis is not convincing because of insufficient data. The authors should provide clearer or additional data to clarify whether SIRT1 or its target TRIM23 affects adipogenesis. Otherwise, the authors need to modify their conclusion about Yak’s preadipocyte in the title and abstract of the present manuscript.
1. Overall, the figure data do not show protein analysis. Antibodies to Yak may not be commercially available but are recommended to show histone H3 acetylation, SIRT1, and TRIM3 protein levels in knockdown and overexpression conditions.
2. Authors should provide a rationale in this manuscript for how TRIM23, an important target gene of SIRT1, was selected among various up or down-regulated target genes.
3. It will be important to further analyze whether SIRT1 regulates important adipocyte-specific genes including adipogenesis, lipolysis, and lipogenesis in a histone acetylation-dependent manner.
4. Most studies on SIRT1 and adipogenesis have been conducted in murine and human models. The epigenetic role of SIRT1 in adipocyte differentiation has been well-studied. The authors concluded general biological SIRT1 roles in adipogenesis. There is no reason or benefit to experimenting with yak unless there is a special biological understanding of yak or research on its meat.
Round 2
Reviewer 3 Report
The limitations of the study were well described in this manuscript. I accept publication in ijms.